# Exopolysaccharide from *Lactobacillus plantarum* HY7714 Protects against Skin Aging through Skin–Gut Axis Communication

**DOI:** 10.3390/molecules26061651

**Published:** 2021-03-16

**Authors:** Kippeum Lee, Hyeon Ji Kim, Soo A Kim, Soo-Dong Park, Jae-Jung Shim, Jung-Lyoul Lee

**Affiliations:** R&BD Center, Korea Yakult Co Ltd., 22, Giheungdanji-ro 24 beon-gil, Giheung-gu, Gyeonggi-do 17086, Korea; joy4917@hanmail.net (K.L.); hyeonjk@re.yakult.co.kr (H.J.K.); freebi7@re.yakult.co.kr (S.A.K.); soodpark@re.yakult.co.kr (S.-D.P.); jjshim@re.yakult.co.kr (J.-J.S.)

**Keywords:** exopolysaccharide, lactic acid bacteria, skin aging, photoaging, Caco-2 cells, HS68 cells

## Abstract

Skin aging occurs inevitably as a natural result of physiological changes over time. In particular, solar exposure of the skin accounts for up to 90% of skin damage. Numerous studies have examined the ability of dietary constituents to prevent skin aging, and recent research has emphasized the role of functional probiotics in intestinal function and skin aging. However, the mechanism of the interactions between aging and probiotics has not been elucidated yet. The aim of this study was to determine the role of exopolysaccharides (EPS) produced by lactic acid bacteria (LAB) identified as *Lactobacillus plantarum* HY7714 in regulating tight junctions in intestinal epithelial cells and increasing moisture retention in human dermal fibroblasts cells. We observed that HY7714 EPS controlled intestinal tight junctions in Caco-2 cells by upregulating the genes encoding occludin-1 (OCL-1) and zonula occluden-1 (ZO-1). In addition, HY7714 EPS effectively improved UVB-induced cytotoxicity and hydration capacity in HS68 cells by downregulating production of metalloproteinases (MMPs) and reactive oxygen species (ROS). In summary, HY7714 EPS is an effective anti-aging molecule in skin and may have therapeutic potential against skin diseases and UVB-induced damage. Therefore, HY7714 EPS serves as a functional substance in skin–gut axis communication.

## 1. Introduction

Skin aging is a complex process caused by both extrinsic and intrinsic factors [1]. Intrinsic aging is a natural process associated with physiological, hormones, genetic, and cellular metabolic changes [2]. By contrast, extrinsic aging is the result of various environmental causes, including air pollution and exposure to sunlight. Ultraviolet (UV) irradiation, a major cause of external changes to the skin, increases production of intracellular reactive oxygen species (ROS) and pro-inflammatory cytokines [3]. ROS production caused by exposure to high UV radiation is closely related to oxidative stress in dermal cells [4]. In addition, excessive ROS can indirectly generate DNA changes and membrane imbalances, resulting in a decrease in collagen synthesis. This oxidative stress is responsible for skin aging and disease development [5]. Because skin is the most important and largest surface barrier in the body, it is essential to prevent UV-induced injury in the reconstituted human epidermis.

Metalloproteinases (MMPs), one the most important protease families, can modulate expression of collagen genes in the extracellular matrix (ECM) and is a major component of the cellular microenvironment [6]. ECM, a highly dynamic structure in the skin, is influenced by the external environment and continuously undergoes remodeling processes, such as regeneration and degradation [7]. For instance, recent work showed that UVB radiation causes a molecular chain reaction through upregulation of MMPs expression in the dermis and epidermis [8]. Furthermore, UVB irradiation also promotes cutaneous inflammation [9]. In particular, UVB-induced secretion of metalloproteinases-1 (MMP1) promotes inflammation in oxidatively stressed environments. In addition, transient expression of MMP1, MMP3, and MMP13 correlates with ECM degradation, cell adhesion, tissue remodeling, and proliferation. [10]. UVB exposure is also associated with secretion of cytokines, including Interleukin 1 beta (IL-1β) and tumor necrosis factor-alpha (TNFα), in skin fibroblasts [11]. TNFα plays a critical role in photoaging by activating the expression of elastase and collagenase, which damage the skin [11]. Treatment with bioactive natural compounds has the potential to control these pathways and prevent photo-damage. As awareness of the harmful effects of chronic UVB exposure has grown, an increasing amount of research attention has been devoted molecules with anti-aging effects on the skin [12].

Skin aging is also related to loss of skin moisture. The major glycosaminoglycan hyaluronic acid (HA), an ECM component in skin dermal tissue, is involved in aspects of skin health such as hydration, cell regeneration, development, and wound healing [13]. Due to its net anionic charge and nonsulfated glycosaminoglycan structure, HA promotes water capture and regulates tissue hydration. HA is synthesized directly in the ECM by hyaluronan synthase (HAS1, HAS2, and HAS3) [14]. Reductions in the levels of hyaluronan acid synthase (HAS) enzymes are associated with downregulation of HA, implying that these proteins are primarily responsible for HA synthesis. Therefore, to prevent loss of skin moisture, it is important to increase HAS expression in skin fibroblasts. 

Exopolysaccharide (EPS) consists of long-chain polysaccharides with various branches and repeating units of sugars or sugar derivatives [15]. Recent studies showed that EPS from marine microorganisms has potential applications as a therapeutic food [12,16]. In particular, natural polysaccharides produced by lactic acid bacteria (LAB) possess many biological activities, including immunomodulatory and antioxidant activities, and could be used as functional antioxidants [17]. Moreover, the functionality and composition of EPS differs even among LABs from the same species. EPS biodiversity can result from different combinations of sugar biosynthesis pathways and genetic variability among strains [18].

Skin health is associated with homeostasis of tight junctions in the intestine through skin–gut axis communication [19]. Probiotics, including *Lactobacillus* sp. and *Bifidobacterium* sp., play important roles in cosmetics and aging [20]. These microbes exist in the human gastrointestinal tract and have beneficial anti-aging effects on the skin, but the interaction has not yet been completely elucidated. In a previous study, we examined the effects of *Lactobacillus plantarum* HY7714 (HY7714), a functional probiotic approved by the Korea Food and Drug Administration (KFDA), regarding skin hydration [21]. However, it remains unclear which molecule of HY7714 affects skin health. Hence, in this study, we sought to characterize the effects of HY7714 EPS on human intestinal adenocarcinoma cells (Caco-2) and human dermal fibroblasts (HS68). Briefly, HY7714 EPS was treated on TNFα- damaged Caco-2 cell, in order to prove the protective effect on skin aging of HY7714 EPS through intestinal adhesion regulation. Caco-2 cell lines are most extensively used as a model of the intestinal barrier studies [22]. Caco-2 tight junctions between cells serve as models of human intestinal absorption and natural compound transport across monolayers [23]. In addition, the HS68 cell line is the most representative dermal cell used in skin research, which are related to skin elasticity and integrity through MMP related to collagen synthesis. To establish aged skin cell model, HS68 cells were irradiated with UVB and then treated with HY7714 EPS.

## 2. Results

### 2.1. Purification of EPS from Lactobacillus plantarum HY7714

*L. plantarum* HY7714, previously isolated from the breast milk of healthy women, was cultured in de Man–Rogosa–Sharpe broth (BD, San Diego, CA, USA) at 37 °C for 20 h. As shown in Appendix A, HY7714 produced an opaque and thickly spread EPS, whereas HY7711 EPS did not (Appendix A). Recent research reported that most of lactic acid bacteria can produce EPS, surface carbohydrate polymers with diverse biological functions [16,24]. Lactic acid bacteria HY7714 is granted the status of GRAS and is also known to be the suitable candidates for the production of functional EPS and derivative carbohydrates. Thus, to purify and separate the EPS, ethanol precipitation and TCA treatment was performed on the supernatant, yielding 2 g lyophilized EPS per liter of culture. 

### 2.2. Molecular Weight and Monosaccharide Composition Estimation of HY7714 EPS

In this study, we determined the molecular weight of EPS using a GPC analysis system (Figure 1). We observed five fractions of different molecular weights in the range of 221 to 79,967 Da (HY7714 EPS) and 236 to 56,899 Da (HY7711 EPS). In particular, the largest molecular component (79,967 Da) accounted for 29.1% of HY7714 EPS, whereas the largest component of HY7711 EPS was smaller (56,899 Da) and accounted for 24.5% of the total mass. Next, we performed HPLC to determine the sugar composition (Appendix A). In terms of monosaccharide units, HY7714 EPS was composed of ribose, glucose, and mannose in a ratio of 4.0:1.5:1.0, whereas HY7711 consisted of the same three sugars in the ratio 1.5:2.0:1.0. The presence of different monosaccharide moieties suggests that EPS of HY7714 and HY7711 are hetero-polysaccharides. 

### 2.3. Effect of HY7714 EPS Treatment on Tight Junctions and Inflammatory Cytokines in Intestinal Cells

Tight junctions are core structures that play critical roles in the barrier function of epithelial cells. The scaffold proteins zonula occludens-1 (ZO-1) and surface-expressed protein occludin-1 (OCL-1) are essential structural molecules of tight junctions that regulate the permeability of the barrier. Pro-inflammatory cytokines including IL-6 and IL-1β trigger intestinal inflammation, so that tight junctions are permeable. In addition, pro-inflammatory TNFα is a crucial controller of inflammatory processes, resulting in production of a variety of MMPs such as MMP1 and MMP2 [25]. To confirm that EPS plays a major role in the effects of *L. plantarum* HY7714 on skin health via the gut–skin axis, we isolated EPS in the same manner as from *L. plantarum* HY7711 and evaluated its effect on the regulation of intestinal tight junctions in Caco-2 cells. We compared the levels of mRNAs encoding pro-inflammatory cytokines and MMPs between cells treated with 1 µg/mL EPS and cells treated with 5 × 10^6^ CFU HY7714, corresponding to production of 1 µg/mL EPS. As shown in Figure 2A,B, the mRNA level of *ZO-1* and *OCL-1* decreased by 78.6% and 68.1%, respectively, upon TNFα treatment. The mRNA level of *OCL-1* was restored to 79.9% and 104.6% following treatment with HY7714 and HY7714 EPS, respectively, and to 83.0% and 84.2% following treatment with HY7711 and HY7711 EPS, respectively. The mRNA expression levels of ZO-1 were recovered upto 93.5% and 93.4% in HY7714 and HY7714 EPS-treated cells, respectively. Also, those of HY7711 and HY7711 EPS-treated Caco-2 cells were increased up to 98.0 and 94.9%, respectively. These results indicate that lactobacilli HY7711 and HY7714 and their respective EPS have a similar effect on intercellular junction and permeability of intestinal cells. Meanwhile, treatment with HY7714 EPS inhibited the increase in expression of *IL-1β* and *IL-6* mRNA in TNFα-treated Caco-2 cells (Figure 2C,D). The regulation of tight junctions of HY7714 EPS resulted in the most effective reduction in inflammatory cytokine levels such as IL-6 and IL-1β compared to lactic acid bacteria or other types of EPS. The *MMP1* mRNA level was elevated in TNFα-treated cells but significantly reduced in cells treated with HY7714 or HY7714 EPS. Finally, the *MMP2* mRNA level in TNFα-treated cells decreased much more when the cells were subsequently treated with HY7714 EPS than when treated with HY7714, HY7711, or HY7711 EPS (Figure 2E,F). Our data suggest that HY7714 and its EPS are highly effective in lowering the gene level of MMPs than HY7711 and HY7711 EPS. The difference in these functional effects may be related to composition and structure of the EPS produced by specific strain. Therefore, HY7711 was used as another functional control that exhibited different effects from HY7714 despite belonging to the same strain as *L. plantarum* HY7714.

### 2.4. Effect of HY7714 EPS Treatment on Gene Expression of MMPs, HAS1, and COLa1 

A hallmark of skin photoaging is degradation of collagen via MMPs expressed in dermal fibroblasts, decreasing the structural integrity of the collagenous ECM. The prototypical MMP1, collagenase 1, has been implicated in melanoma and premature skin aging. HA is a glycosaminoglycan that is synthesized directly into the ECM by HAS. HAS is an important enzyme because HA serves a structural function by binding water, which in turn supports volume expansion and elasticity in the skin. Finally, collagen is an important ECM protein, and type 1 collagen (COLa1) accounts for more than 80% of the total collagen in adult human dermis. Hence, we sought to determine whether HY7714 EPS ameliorates MMP1 secretion in HS68 cells. According to Figure 3, MMP1 secretion in HY7714-treated HS68 cells was significantly decreased, to a much greater extent than in HY7711 EPS-treated cells. Next, we analyzed the mRNA expression of *MMP1* and *HAS1* in dermal fibroblasts by quantitative real-time PCR. The levels of both mRNAs were decreased by HY7714 EPS treatment, whereas HY7711 EPS-treated cells exhibited no significant difference. In western blotting, the level of COLa1 protein was increased by HY7714 EPS in a dose-dependent fashion. The MMP1 level in HS68 cells was decreased by treatment with ≥1 µg/mL HY7714 EPS, but we observed no significant effect at 0.1 µg/mL.

### 2.5. Effect of HY7714 EPS on UVB-Protective Activity and Intracellular ROS Production

Next, we investigated the cytotoxic effect of HY7714 EPS on HS68 cells by MTT assay. As shown in Figure 4A, cell viability did not decrease when cells were treated for 24 h with EPS concentrations from 0.01 to 5 μg/mL. The survival of HS68 cells was reduced to 80.0% at a dose of 10 μg/mL EPS. To investigate the protective effect of HY7714 EPS against UVB-induced damage, we irradiated HS68 cells with 30 mJ/cm^2^ UVB. In Figure 4B, HY7714 EPS treatment decreased UVB-induced cytotoxicity. The viability of UVB-irradiated HS68 cells decreased by 72.1%, whereas the viabilities of cells treated with 0.1, 1, and 5 μg/mL EPS were 70.8%, 91.1%, 91.8%, and 110.0%, respectively. Thus, HY7714 EPS protected HS68 fibroblasts against UVB-induced damage.

ROS play crucial roles in metabolic homeostasis and oxidative signaling regulation in dermal cells. Overexposure to UVB leads to intercellular signaling by generating ROS, which accelerates the reduction in collagen synthesis in dermal fibroblasts. In this study, we detected intracellular ROS levels by DCFH-DA fluorescence staining. As shown in Figure 4C,D, 30 mJ/cm^2^ UVB irradiation increased ROS production in HS68 cells to 171.4%, but HY7714 EPS decreased UVB-induced ROS production in a dose-dependent manner. Thus, HY7714 EPS treatment may have antioxidant effects in UVB-irradiated skin fibroblasts.

### 2.6. Effect of HY7714 EPS Treatment on UVB-Induced ECM Degradation in HS68 Cells

UVB radiation promotes DNA damage indirectly by inducing oxidative stress and ROS generation, leading to photoaging [26]. In particular, UVB exposure induces synthesis of MMPs, and can thus affect skin tissue function and structure. To examine the effect of HY7714 EPS on UVB-induced damage, we UVB-irradiated HS68 cells with or without EPS and monitored the mRNA levels of *MMP1*, *MMP3*, *HAS1*, *HAS2*, *HAS3*, and *SPT1* by qRT-PCR. As shown in Figure 5A,B, the levels of *MMP1* and *MMP3* mRNA were higher in the UVB radiation group than in the control, but a low concentration of HY7714 EPS (0.1 μg/mL) markedly decreased the levels of both mRNAs. In addition, as shown in Figure 5C, UVB damage also decreased the level of *SPT1* mRNA by 0.4-fold. While, HY7714 EPS treatment significantly increased even at the concentration of 0.1 μg/mL, but there was no concentration-dependent effect. In addition, the mRNA levels of *HAS1*, *HAS2*, and *HAS3*, which regulate the synthesis of hyaluronic acid, were increased by 14.1-fold, 1.1-fold, and 4.7-fold in 5 μg/mL in UVB-irradiated HS68 cells treated with HY7714 EPS (Figure 5D,E).

To evaluate the effect of HY7714 EPS on MMP1 production in irradiated HS68 cells, we exposed the cells to UVB (30 mJ/cm^2^) and detected secreted MMP1 by ELISA. A dose of 5 µg/mL HY7714 EPS attenuated UVB-induced MMP1 secretion by 74.6% (Figure 6A). In addition, as shown in Figure 6B, HY7714 EPS treatment rescued the UVB-induced downregulation of HA production in these cells. Finally, western blotting revealed that EPS treatment decreased expression of MMP1 and MMP13 in irradiated cells (Figure 6C).

### 2.7. Effect of HY7714 EPS Treatment on Inflammatory Cytokine Production in HS68 Cells

Photoaging upregulates the levels of inflammation-associated markers, such as TNFα. Consistent with this, production of IL-1β, IL-6, and IL-13 is upregulated following UVB exposure. These cytokines play important roles in photoaging and photo-damage by driving apoptosis and lymphocyte activation in the skin. Figure 7 shows that UVB exposure significantly increased the mRNA levels of *TNFα*, *IL-1*β, *IL-6*, and *IL-13* in HS68 cells. However, HY7714 EPS treatment decreased the mRNA levels of *TNFα*, *IL-6*, and *IL-13* in a dose-dependent manner. In particular, in cells treated with 5 µg/mL HY17714 EPS, TNFα, *IL-1β*, *IL-6*, and *IL-13* decreased by 70.0%, 42.6%, 37.4%, and 33.7% relative to the UVB-irradiated group. Finally, secretion level of pro-inflammatory cytokines was investigated using ELISA kit. TNFα in HS68 cells increased after UVB exposure, but was slightly decreased by HY7714 EPS treatment but there was no significant difference (Figure 7E). While, IL-1β and IL-6 in HS68 cells increased after UVB exposure but were attenuated significantly by HY7714 EPS (Figure 7F,G). These results indicate that HY7714 EPS suppresses UVBS-induced release of pro-pro-inflammatory cytokines.

## 3. Discussion

Gut and skin, densely vascularized organs with important immune roles, are intimately related in purpose and function [27]. Recent work showed that gut and skin share a number of crucial characteristics, with the diet and gut microbiota affecting the skin [28]. In particular, healthy aging is closely connected to the gut-skin communication: because the skin and intestine are the primary interfaces to the external environment, the maintenance of physiological homeostasis in both organs is essential [27]. The mechanisms underlying this positive microbial communication have not yet been elucidated, but some are immune-based. In addition, probiotics act as positive modulators in gut health and oxidative immune regulation [29]. 

Probiotics that affect skin health have been identified, especially among lactobacilli [30]. For instance, a clinical study reported that *L. lactis* strain H61 improved the skin elasticity of middle aged-women [31]. Another study showed that *L. rhamnosus* has a potential to improve skin hydration [32]. Previous work by our group showed that *L. plantarum* HY7714 increases skin moisture and elasticity and decreases the wrinkle depth in human subjects aged 41–59 [21]. However, the mechanisms underlying these effects have not been fully elucidated. One attractive hypothesis is that polysaccharides produced by LAB are important factors in skin health. In this study, we investigated whether certain HY7714 polysaccharides could serve as functional substances that act on the gut–skin axis to change the properties of dermal cells.

Most lactobacillus EPS molecules are heteropolymers consisting of repeated copies of oligosaccharide units [15]. EPS, which are produced either intracellularly or extracellularly, contribute to biological activity via specific composition, size, and branching structure, which can differ even among members of the same LAB strain [33]. Given that microbial EPS has various effects of improving biological and cosmetic functions, it is very important to explore the biodiversity of the naturally derived LAB strains that produce high levels of EPS [34]. The yield of EPS synthesized by Lactobacill can be affected by composition of the medium and growth conditions [35]. In addition, EPS properties in slime form can negatively affect the gradual loss of probiotics, so efforts should be made to standardize culture methods to maintain quality and purity of EPS. When HY7714 probiotics are cultured, slippery substances containing EPS are secreted into the surroundings, making centrifugation difficult. On the other hand, another strain of *L. plantarum*, HY7711, can be centrifuged because it produces a different kind of EPS. We noted these differences in microbial properties, and then compared and analyzed the specific structure of HY7714 and HY7711 EPS. HPLC analyses performed under two different column conditions revealed that HY7714 and HY7711 EPS have different proportions of the same three monosaccharides, ribose:mannose:glucose, in a ratio of 4.0:1.5:1.0 (HY7714) or 1.5:2.0:1.0 (HY7711), respectively. The molecular species in HY7714 EPS and HY7711 EPS, which account for 30% of the total mass, was about 80 kDa and 57 kDa, respectively. Thus, we can infer that HY7714 EPS has a specific phenotype due to the inherent structural characteristics of complex polysaccharide composition. These findings were corroborated by the fact that HY7714 also produced sugar units with a higher molecular weight, as revealed by GPC analysis, indicating that its EPS is larger.

The most valuable application of lactobacillus EPS to date has been improvement of the texture and mouthfeel of fermented milk products [36]. According to a recent hypothesis, EPS can remain for long periods of time in the gastrointestinal tract, improving colonization by probiotics [37]. Consistent with this, in HY7714 EPS-treated Caco-2 cells, mRNA levels of *ZO-1* and *OCL-1* were higher than in cells treated with TNFα alone, which increases intestinal tight junction permeability. This elevated permeability is accompanied by a reduction in *ZO-1* and *OCL-1* levels, resulting in leakage of pro-inflammatory cytokines into blood vessels and other tissues [38]. According to our data, HY7714 EPS decreased the secretion of *IL-1β* and *IL-6* in TNFα-treated cells and restored them to their usual levels. According to recent studies, a few types of matrix metalloproteinase (MMP) play vital roles in the development of inflammation in intestinal epithelial cells. Multiple studies have reported that IL-1β and TNFα increase expression of MMP [39,40]. At the same time, MMPs play a role in promoting ECM degradation following UVB damage in skin dermis and epidermal cells. Notably in this regard, the gut environment induces redistribution of skin homeostasis after UV irradiation [41,42].

UV exposure damages the structure and function of skin, and has therefore been implicated in sunburn, immunity, cancer, and photoaging [43]. UV light is composed of UVC (200–280 nm), UVB (280–315 nm), and UVA (315–400 nm) [26]. In particular, UVB irradiation promotes the production of ROS and induces the overexpression of MMP1, MMP3, and MMP9 in human fibroblasts, resulting in the destruction of collagen and ultimately to wrinkle formation [44,45]. In this study, we investigated whether the protective effects of HY7714 EPS against UVB irradiation are governed by its ability to protect against oxidative stress and MMP expression. We found that UV-induced oxidative damage and induction of MMPs in HS68 cells were significantly decreased by HY7714 EPS treatment. Therefore, HY7714 EPS has the potential to alleviate UVB damage of dermal connective tissue, a collagenous ECM. MMPs, which degrade cutaneous protein, are upregulated by UVB irradiation, resulting in a loss of elasticity and promotion of wrinkle formation. The MMP can be classified into several subgroups: collagenases (MMP-1, MMP-8, MMP-13), gelatinases (MMP-2, MMP-9), and stromelysins (MMP-3, MMP-10, MMP-11). As our data show, HY7714 EPS decreased the mRNA levels of *MMP1* and *MMP3*, which are upregulated by UVB exposure, and significantly decreased the protein level of MMP13. By contrast, HY7711 EPS did not inhibit MMP activation. In summary, HY7714 EPS protects the skin by inhibiting ECM degradation. 

HA present in skin cells is synthesized by hyaluronic acid synthase, a membrane-bound enzyme expressed by keratinocytes and dermal fibroblasts. HAS produces HA of varying lengths, with HAS1 and HAS2 mainly synthesizing large units of HA polymers [46]. Meanwhile, decomposition of HA in skin dermal cells following UVB exposure causes wrinkles and loss of elasticity and moisture. According to our data, HY7714 EPS significantly increased the mRNA expression levels of control *HAS1* and *HAS2* in UVB-exposed HS68 cells. It is already known that upregulation of *HAS2* mRNA expression plays a pivotal role in HA synthesis. Moreover, we found that HY7714 EPS upregulated the mRNA level of serine palmitoyl transferase (SPT1), a major enzyme involved in the biosynthesis of ceramide in skin cells. Thus, HY7714 EPS may promote synthesis of HA and ceramide.

UVB exposure also contributes to the inflammatory response by driving generation of ROS and secretion of mediators such as inflammatory cytokines. Subsequently, ROS accumulation mediates the UVB-induced expression of MMP1. We observed that UVB irradiation rapidly increased ROS levels, whereas HY7714 EPS inhibited ROS in UVB-exposed cells in a dose-dependent manner. The increase in pro-inflammatory cytokines caused by UVB exposure drives photoaging in skin dermal cells. Indeed, UVB irradiation in HS68 cells promoted acute inflammation, resulting in stimulation of IL-1β, IL-6, IL-13, and tumor necrosis factor (TNFα). However, HY7714 EPS significantly decreased the levels of interleukin in a dose-dependent manner. Taken together, these findings suggest that HY7714 EPS regulates photoaging by attenuating ROS regulation and secretion of pro-inflammatory cytokines. 

In summary, our findings reveal a novel effect of HY7714 EPS, namely, improving the function of intestinal tight junctions in human-derived Caco-2 cells. In addition, HY7714 EPS increased the mRNA levels of MMPs, which can affect damaged skin cells, thereby decreasing the expression of pro-inflammatory cytokines. Moreover, HY7714 EPS attenuated ECM degradation by promoting HA synthesis and inhibiting MMP expression in dermal HS68 cells. Therefore, HY7714 has the potential to benefit skin health through the microbiome–skin–gut axis. These findings reveal that EPS from HY7714 is a biologically effective substance with potential value as a cosmetic or nutraceutical. 

## 4. Materials and Methods

### 4.1. EPS Isolation and Purification

EPS was obtained from supernatant of *L. plantarum* by ethanol precipitation. Briefly, culture media of *L. plantarum* HY7714 was separated by centrifugation (8000× *g* for 20 min at 4 °C). Chilled ethanol was slowly added to the supernatant and incubated for 24 h at 4°C; the precipitate was recovered by centrifugation (8000× *g* for 20 min at 4 °C). For purification, 4% (*v*/*v*) trichloroacetic acid was added, and the sample was incubated 4°C for 2 h to remove proteins. The supernatant was filtered through a 0.45 μm cellulose nitrate filter (Millipore, Bangalore, India) to remove remaining protein, and the remaining ethanol was evaporated off, yielding purified EPS. EPS was at 4 °C prior to chemical and physical analyses.

### 4.2. Molecular Weight Analysis of EPS

Molecular weight of the purified EPS was checked on a gel permeation chromatograph (GPC) equipped with TSK gel guard PWXL, TSK gel GMPWXL, and TSK gel G-2500 PWXL (7.8 × 300 mm) columns (TOSOH, Tokyo, Japan) in conjunction with a refractive index detector. Two hundred microliters of 3 mg/mL EPS was injected and eluted with 0.1 M NaNO_3_ solution at 40 °C at a flow rate of 1.0 mL/min. Data were detected and processed using the EcoSEC software (Tosoh Bioscience, Tokyo, Japan). Molecular weights were calculated using pullulan standards (Sigma-Aldrich, St. Louis, MO, USA). 

### 4.3. Determination of Monosaccharide Composition of EPS

Five milligrams of purified EPS and standards were dissolved in 1.6 mL of 2 N H_2_SO_4_ at 100 °C for 6 h. The EPS solution was neutralized with 3.3 mL of 1 N NaOH solution. The remaining protein and fat in the resultant lysates was removed by addition of 12% (*v/v*) Biggs-Szijarto solution (Sigma-Aldrich), centrifugation (13,000× *g*, 10 min, 4°C), and filtration through a 0.45 μm regenerated cellulose (RC) filter (Whatman, Kent, England). To analyze the monosaccharide composition, 20 µL samples were analyzed by high-performance liquid chromatography (HPLC) (1260 Infinity, Agilent, Santa Clara, CA, USA) equipped with a Shodex Asahipak NH2P-50 4E column (4.6 mm × 250 mm, 5 μm) and eluted with 72% acetonitrile at a flow rate of 0.8 mL/min. The separated components were monitored using a refractive index (RI) detector. To crosscheck, samples were analyzed on an Agilent 1260 Infinity instrument equipped with an Imtakt Unison UK-Amino (UKA66, 4.6 mm × 250 mm, 3 μm, Imtakt, Tokyo, Japan) column and eluted with methanol at a flow rate of 1.2 mL/min. The separated components were also monitored using a RI detector. The column was calibrated with a molecular mass standard, and a standard curve was established for each column condition.

### 4.4. Cell Culture

HS68 (CRL-1635) human dermal fibroblasts obtained from American Type Culture Collection (Manassas, VA, USA) were maintained in Dulbecco’s modified Eagle’s medium (DMEM) with 1% penicillin–streptomycin (P/S) and 10% heat-inactivated fetal bovine serum (FBS) at 37 °C in humidified air containing 5% CO_2_. The cells were seeded in 12-well plates (5 × 10^4^ cells/well) for 24 h and used between passage numbers 6 and 15.

Caco-2 human intestinal adenocarcinoma cells were purchased from the Korean Cell Line Bank (Seoul, Korea). Cells were cultured in DMEM essential medium supplemented with 1% P/S and 10% FBS at 37 °C in humidified air containing 5% CO_2_. Cells were fully differentiated for 21 days, and growth medium was refreshed every 2 days.

### 4.5. EPS Treatment and UVB Irradiation

To investigate the protective effect of EPS, HS68 cells were cultured in growth medium for 24 h to reach 80% confluence. The cells were pretreated with various concentrations (0.1, 1, or 5 μg/mL) of EPS in serum-free medium for 24 h. After the medium was replaced with PBS, the cells were exposed to UVB irradiation at 30 mJ/cm^2^ using an Ultraviolet Cross-linker (UVP, Upland, CA, USA). After UVB exposure, the cells were immediately treated with EPS in serum-free medium for an additional 24 h. 

### 4.6. Cell Viability

To determine the appropriate concentrations of EPS for use in subsequent investigations, we performed cell viability assays using 3-(4,5-dimethylthiazol-2-yl)-2,5-diphenyltetrazolium bromide (MTT). HS68 fibroblasts were seeded in 96-well plates at 1 × 10^4^ cells/well and incubated overnight. Stock solutions of EPS were prepared in distilled water. The cells were treated with EPS (0, 0.01, 0.1, 1, 5, or 10 μg/mL) for 24 h; non-treated cells were used as negative controls. Twenty microliters of a 5 mg/mL MTT solution was added to each well, and the cells were incubated for a further 4 h, leading to the formation of purple formazan crystals. After the MTT-containing medium was removed, 100 μL DMSO was added to elute the formazan crystals. The optical density of the formazan solution, which is associated with the enzyme activity and the number of viable cells, was quantified at 570 nm on a BioTek ELISA reader (Winooski, VT, USA).

### 4.7. Measurement of MMP1, HA, TNFα, IL-1β and IL-6

Cell culture medium was collected, and MMP1 was quantified using commercial ELISA/calorimetric assay kits (CUSABIO, Houston, TX, USA; CSB-E04672h). HA in medium was determined using ELISA/calorimetric assay kits (CSB-E04805h). To induce MMP1 overexpression, HS68 cells were treated with 100 ng/mL TNFα with or without EPS. Cytokines including TNFα, IL-1β and IL-6 in medium were detected using commercial ELISA/calorimetric assay kits (BD OptEIA™, BD-555212, BD-557953, BD-555220).

### 4.8. Measurement of Intracellular ROS

Cells were cultured and pretreated with EPS (0.1, 1, or 5 μg/mL) for 24 h, washed twice in PBS, and then exposed to 30 mJ/cm2 UVB. After that, the cells were stained with 10 μM DCFH-DA and analyzed using a Axiovert 200M fluorescence spectrophotometer (Zeiss, Oberkochen, Germany). Intercellular ROS levels, visualized by DCFH-DA fluorescence, were measured on a BioTek Synergy H1 hybrid microplate reader (BioTek, Winooski, VT, USA) with excitation and emission wavelengths of 485 nm and 530 nm, respectively. 

### 4.9. Quantitative Reverse Transcription Polymerase Chain Reaction (qRT-PCR) Analysis

RNA was isolated from cells using the TRIzol reagent (Invitrogen, Carlsbad, CA, USA). cDNA was synthesized from 2 μg RNA on a thermal cycler (Bio-Rad, Hercules, CA, USA) using Maxime RT PreMix (iNtRON Biotechnology, Seongnam, Korea); the reaction ran for 60 min. The cDNA was analyzed by qPCR (Applied Biosystems, Carlsbad, CA, USA) using the TaqMan Probe-Based Gene Expression analysis system in combination with TaqMan Gene Expression Master Mix containing ROX (Applied Biosystems). Quantification of *MMP1* (Hs00899658_m1), *MMP2* (Hs01548727_m1), *MMP3* (Hs00968306_g1), *HAS1* (Hs00987418_m1), *HAS2* (Hs00193435_m1), *SPT1* (Hs00370543_m1), *TNFα* (Hs99999043_m1), *IL-1β* (Hs01555410_m1), *IL-6* (Hs00174131_m1), *IL-13* (Hs00174379_m1), *ZO-1* (Hs01551861_m1), *OCL1* (Hs00170162_m1), and *GAPDH* (Hs03929097_g1) transcripts was performed using gene-specific primers. Expression data were normalized against the corresponding level of *GAPDH*. To compare mRNA levels between groups, relative mRNA levels were calculated using the 2^(−ΔΔCT^^)^ method.

### 4.10. Western Blot Analysis

Cells were lysed in lysis buffer (iNtRON Biotechnology, Seoul, Korea), and lysate protein concentrations were quantified using a protein assay kit (Bio-Rad). Equal amounts of protein were subjected to SDS-PAGE and electro-transferred to membranes. The membranes were blocked with 5% skim milk in Tris-buffered saline containing Tween 20 (TBS-T) for 1 h, washed with TBS-T, incubated with primary antibodies overnight at 4°C, and then exposed to horseradish peroxidase-conjugated secondary antibodies. Antibodies targeting MMP1, MMP13, collagen type I alpha 1, (COLa1), and glyceraldehyde 3-phosphate dehydrogenase (GAPDH) were purchased from Cell Signaling Technology (Danvers, MA, USA). 

### 4.11. Statistical Analysis

Data are expressed as means and standard deviations (SDs). Data were analyzed by one-way ANOVA, followed by Duncan’s test (IBM SPSS Statistics Version 20.0, Chicago, IL, USA). Statistical significance was defined as *p* < 0.05 (a > b > c > d).

## 5. Conclusions

We found that EPS from *L. plantarum* HY7714 has the potential to not only protect against UVB-induced photoaging in human dermal cells, but also to regulate tight junctions in human intestinal cells. HY7714 EPS ameliorated the inflammatory response and MMP synthesis in Caco-2 cells, which are related to upregulation of MMPs following UVB exposure in HS68 skin fibroblasts, and thus affected regulation of the skin ECM. In addition, HY7714 EPS consists of a specific ratio of ribose, glucose, and mannose, and exhibits characteristics completely different from EPS in other types of *L. plantarum*. Taken together, our findings suggest that HY7714 EPS might be useful as a cosmetic and functional dietary substance. 

## Figures and Tables

**Figure 1 molecules-26-01651-f001:**
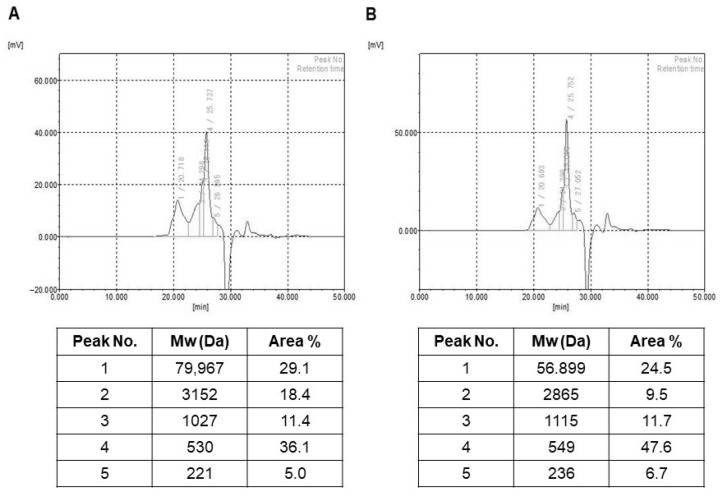
Gel permeation chromatography (GPC) analysis of lactic acid bacteria EPS. GPC chromatogram and peak information of (**A**) *L. plantarum* HY7714 and (**B**) *L. plantarum* HY7711.

**Figure 2 molecules-26-01651-f002:**
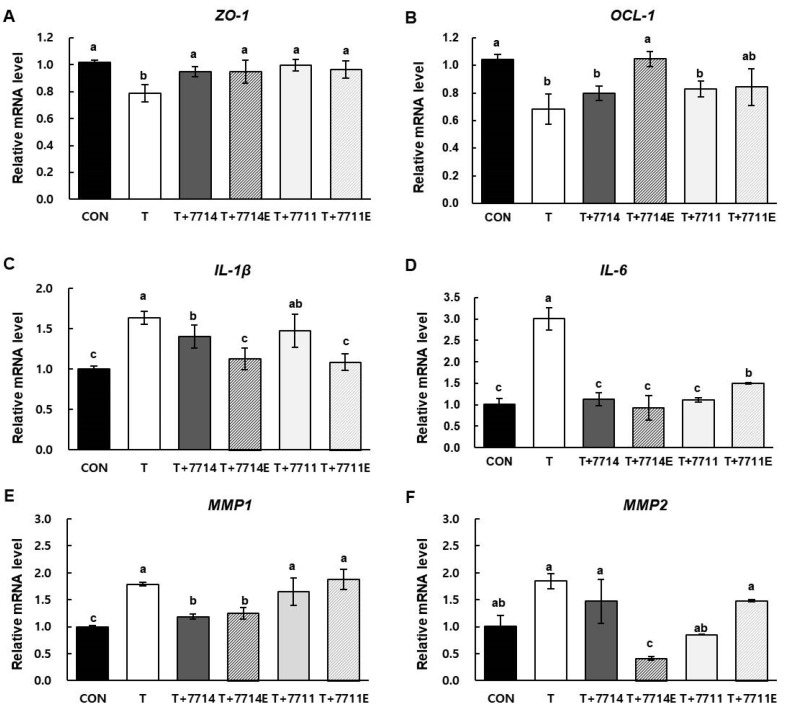
Effect of *L. plantarum* and their EPS on intestinal adhesion and inflammation in Caco-2 cells. Relative mRNA expression levels of (**A**) *ZO-1*, (**B**) *OCL-1*, (**C**) *IL-1β*, (**D**) *IL-6*, (**E**) *MMP1*, and (**F**) *MMP2* in Caco-2 cells were investigated by qPCR and normalized against GAPDH. mRNA data are expressed as means ± SD (*n* = 4), and values labeled with different letters are significantly different, *p* < 0.05 (a > b > c > d). CON, control cells; T, cells treated with 100 ng/mL TNFα; T+7714, cells treated with 100 ng/mL TNFα and 5 × 10^6^ CFU of HY7714; T+7714E, cells treated with 100 ng/mL TNFα with 1 μg/mL HY7714 EPS; T+7711, cells treated with 100 ng/mL TNFα with 5 × 10^6^ CFU of HY7711; T+7711E, cells treated with 100 ng/mL TNFα with 1 μg/mL HY7711 EPS.

**Figure 3 molecules-26-01651-f003:**
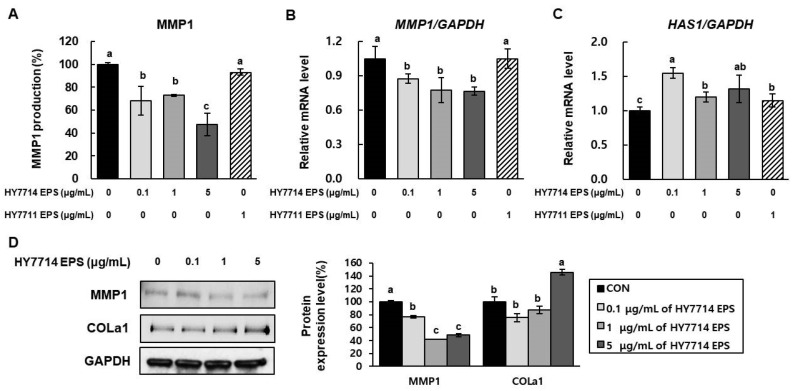
Effect of HY7714 EPS on ECM modulation in HS68 cells. (**A**) The intracellular level of matrix metalloproteinase1 (MMP1) in HS68 cells treated with HY7714 EPS was measured using an MMP1 ELISA kit. (**B**,**C**) Relative mRNA levels of MMP1 and HAS1 in HS68 cells were calculated by qPCR and normalized against GAPDH. (**D**) Western blotting for MMP1 and COLa1 in HS68 cells, and quantification of band density. Data are expressed as means ± SD (*n* = 4), and values labeled with different letters are significantly different, *p* < 0.05 (a > b > c). CON, control cells.

**Figure 4 molecules-26-01651-f004:**
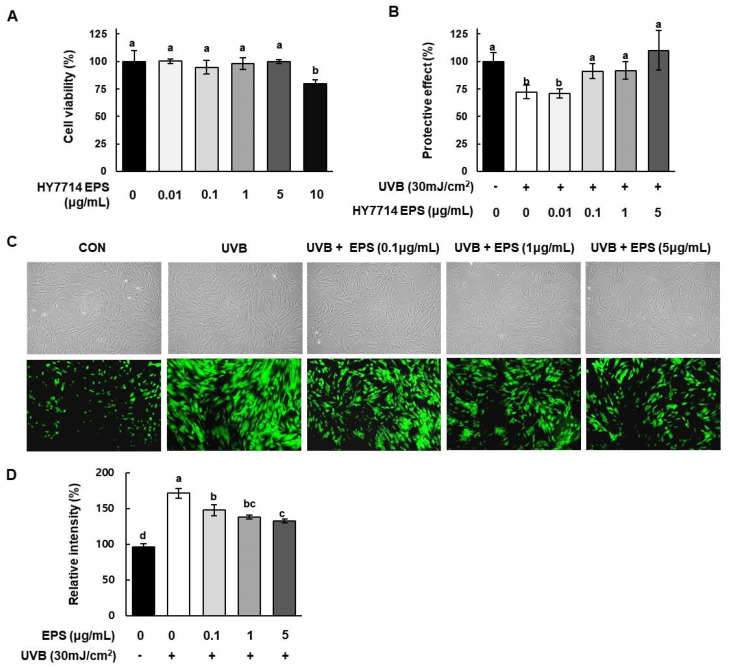
Effects of HY7714 EPS on viability and ROS production of HS68 cells. (**A**) Effect of different concentrations (0.01, 0.1, 1, 5, and 10 μg/mL) of HY7714 EPS on viability of HS68 cells, as determined by MTT assay. (**B**) Cell viability of ultraviolet B (UVB)-induced HS68 cells at different concentrations (0.01, 0.1, 1, 5, and 10 μg/mL) of HY7714 EPS. Cells were irradiated with 30 mJ/m^2^ UVB. (**C**) Effect of HY7714 EPS on the level of ROS, as determined by DCFH-DA fluorescence. Representative image shows fluorescence staining of HS68 cells exposed to the indicated treatments. (**D**) Production of intracellular ROS was quantified using ImageJ. Data are expressed as means ± SD (*n* = 3), and values labeled with different letters are significantly different, *p* < 0.05 (a > b > c > d).

**Figure 5 molecules-26-01651-f005:**
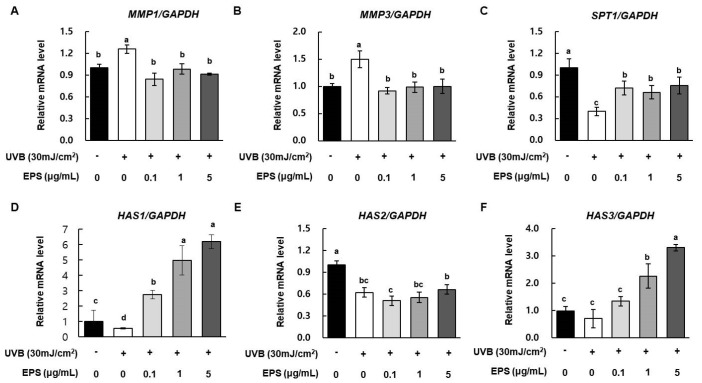
Effect of HY7714 EPS on ECM modulation in UVB-irradiation HS68 cells. Cells were irradiated with 30 mJ/m^2^ UVB and treated with or without HY7714 EPS. Relative mRNA levels of (**A**) MMP1, (**B**) MMP3, (**C**) SPT1, (**D**) HAS1, (**E**) HAS2, and (**F**) HAS3 in HS68 cells were monitored by qPCR and normalized against GAPDH. mRNA data are expressed as means ± SD (*n* = 4), and values labeled with different letters are significantly different, *p* < 0.05 (a > b > c > d).

**Figure 6 molecules-26-01651-f006:**
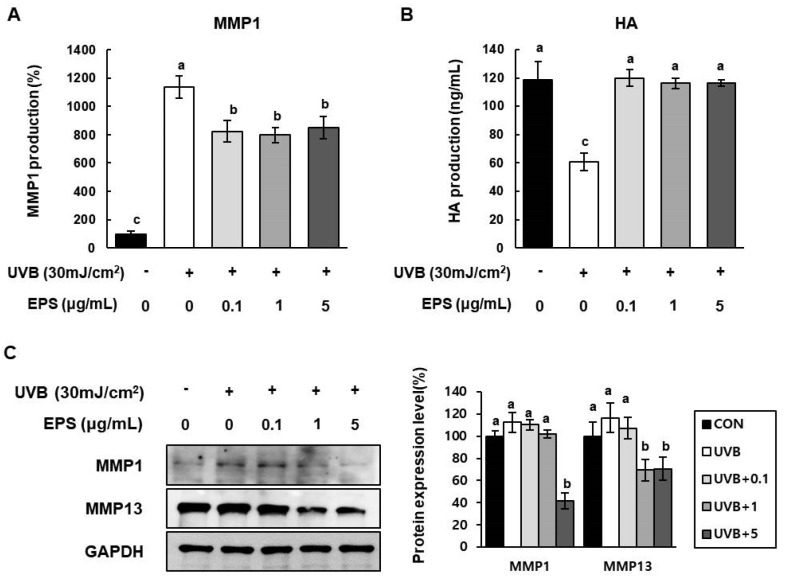
Effect of HY7714 EPS on protein levels of MMPs and intracellular hyaluronic acid (HA) in UVB-irradiated HS68 cells. (**A**) Relative levels of intracellular MMP1 in HS68 cells treated with HY7714 EPS were measured using an MMP1 ELISA kit. MMP1 production was amplified by pre-treatment with 100 ng/mL TNFα for 2 h. (**B**) Intracellular HA in UVB-exposed cells treated with HY7714 EPS was measured using an HA ELISA kit. (**C**) Western blotting for MMP1 and MMP13 in HS68 cells, and quantification of band density. Data are expressed as means ± SD (*n* = 3), and values labeled with different letters are significantly different, *p* < 0.05 (a > b > c).

**Figure 7 molecules-26-01651-f007:**
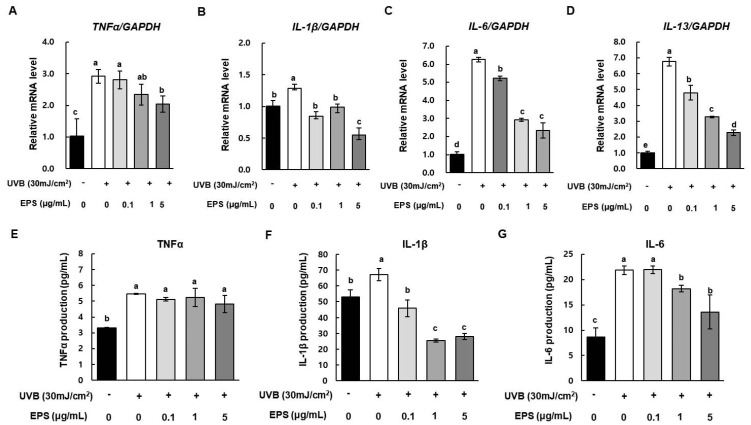
Effect of HY7714 EPS on pro-inflammatory cytokines in UVB-irradiated HS68 cells. (**A**) Relative mRNA expression levels of (**A**) *TNFα*, (**B**) *IL-1β*, (**C**) *IL-6*, and (**D**) *IL-13* in HS68 cells were measured by qPCR and normalized against *GAPDH*. Relative levels of intracellular (**E**) TNFα, (**F**) IL-1β, and (**G**) IL-6 in medium of HS68 cells treated with HY7714 EPS were measured using commercial ELISA kits. Data are expressed as means ± SD (*n* = 4), and values labeled with different letters are significantly different, *p* < 0.05 (a > b > c > d > e).

## Data Availability

Data is contained within the article or Appendix A.

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
