# Peer review of "Exopolysaccharide from Lactobacillus plantarum HY7714 Protects against Skin Aging through Skin–Gut Axis Communication"

_molecules, 2021, doi:10.3390/molecules26061651_

Round 1

Reviewer 1 Report

In this manuscript of "Exopolysaccharide from Lactobacillus Plantarum HY7714 Protects Against Skin Aging Through Skin–gut Axis Communication" authors claimed that HY7714 EPS effectively improved UVB-induced cytotoxicity and hydration capacity in HS68 cells and thus can serve as a functional substance in skin–gut axis communication. A lot of experimental work was done and it is certainly contributing to the current scientific knowledgebase. However, I am puzzled by somewhere in the paper. The following are the questions and comments of this manuscript:

  1. In the introduction part, may authors explain more about why choose Caco-2 and HS68 these two specific cells to conduct experiment?
  2. In 2.1 Purification of EPS from Lactobacillus plantarum HY7714, can authors make some speculations about “opaque and thickly” thing?
  3. In 2.3 line 17-18, why the level of ZO-1 recovered to an average of 95% regardless of strain or source material?
  4. In Fig.2 E, why the T + 7711 bar reached the highest relative mRNA level even higher than T bar? May authors explain clearer about the purpose of using HY7711 in these figures?
  5. In 2.6 line 9-11, can authors try to explain the reason that no statistically significant difference in the SPT1 level in cells treated with 0.1–5 ug/mL EPS? Also, in Fig.6 C, it’s not so clear in MMP13 band, especially the left band.
  6. In the discussion part line 22-23, authors mentioned that EPS can differ even among members of the same LAB strain, so how to detect and ensure the quality and purity of EPS if using it in cosmetic and functional dietary way?

After going through the manuscript, I feel that authors need to revise the paper to reach the level of Molecules.

Author Response

Dear reviewer 1

Thank you for considering our manuscript for publication in Molecules. We are very pleasure to have been given the opportunity to revise our manuscript, “Exopolysaccharide from Lactobacillus plantarum HY7714 protects against skin aging through skin–gut axis communication”. We have addressed the reviewer’s comments point-by-point and made the necessary changes to the manuscript. We hope that the manuscript is now acceptable for publication in Molecules. Please see the attachment.

Sincerely,

Jung-Lyoul Lee, Ph.D

1) In the introduction part, may authors explain more about why choose Caco-2 and HS68 these two specific cells to conduct experiment?

Answer: We appreciated for your suggestion. As reviewer mentioned, we revised it in Introduction.; “Hence, in this study, we sought to characterize the effects of HY7714 EPS on human intestinal adenocarcinoma cells (Caco-2) and human dermal fibroblasts (HS68). Briefly, HY7714 EPS was treated on TNFα- damaged Caco-2 cell, in order to prove the protective effect on skin aging of HY7714 EPS through intestinal adhesion regulation. Caco-2 cell lines are most extensively used as a model of the intestinal barrier studies [1]. Caco-2 tight junctions between cells serve as models of human intestinal absorption and natural compound transport across monolayers [2]. In addition, the HS68 cell line is the most representative dermal cell used in skin research, which are related to skin elasticity and integrity through MMP related to collagen synthesis. To establish aged skin cell model, HS68 cells were irradiated with UVB and then treated with HY7714 EPS.”

2) In 2.1 Purification of EPS from Lactobacillus plantarum HY7714, can authors make some speculations about “opaque and thickly” thing?

Answer: Thank you for your suggestion. Regarding on this, Result 2.1 was added according to reviewer comments.; “As shown in Figure S1, HY7714 produced an opaque and thickly spread EPS, whereas HY7711 EPS did not (Figure S1). Recent research reported that most of lactic acid bacteria can produce EPS, surface carbohydrate polymers with diverse biological functions [3,4]. Lactic acid bacteria HY7714 is granted the status of GRAS and is also known to be the suitable candidates for the production of functional EPS and derivative carbohydrates. Thus, to purify and separate the EPS, ethanol precipitation and TCA treatment was performed on the supernatant, yielding 2 g lyophilized EPS per liter of culture.”

3) In 2.3 line 17-18, why the level of ZO-1 recovered to an average of 95% regardless of strain or source material?

Answer: We described in the Result 2.3.; “The mRNA expression levels of ZO-1 were 93.5% and 93.4% in HY7714 and HY7714 EPS-treated cells, respectively. Also, those of HY7711 and HY7711 EPS-treated Caco-2 cells were 98.0 and 94.9%, respectively, and there was no significant difference from each other. These results indicate that lactobacilli HY7711 and HY7714 and their respective EPS have a similar effect on intercellular junction and permeability of intestinal cells. Meanwhile, treatment with HY7714 EPS inhibited the increase in expression of IL-1β and IL-6 mRNA in TNFα-treated Caco-2 cells (Figure 2C–D). The regulation of tight junctions of HY7714 EPS resulted in the most effective reduction in inflammatory cytokine levels such as IL-6 and IL-1β compared to lactic acid bacteria or other types of EPS.”

4) In Fig.2 E, why the T + 7711 bar reached the highest relative mRNA level even higher than T bar? May authors explain clearer about the purpose of using HY7711 in these figures?

Answer: Thank you for your pointing out. We modified Figure 2E, which means that despite the same L. plantarum as HY7714, HY7711 lactobacilli and its EPS have no effect on inhibiting MMP1 expression. In addition, these differences in effects may be related to the unique properties of the EPS produced by the strain. In order to reduce the reader's misunderstanding, the data was re-sorted from a number of repetitive experiments. And we described in Result 2.3.; “Our data suggest that HY7714 and its EPS are highly effective in lowering the gene level of MMPs than HY7711 and HY7711 EPS. The difference in these functional effects may be related to composition and structure of the EPS produced by specific strain. Therefore, HY7711 was used as another functional control that exhibited different effects from HY7714 despite belonging to the same strain as L. plantarum HY7714.”

5) In 2.6 line 9-11, can authors try to explain the reason that no statistically significant difference in the SPT1 level in cells treated with 0.1–5 ug/mL EPS? Also, in Fig.6 C, it’s not so clear in MMP13 band, especially the left band.

Answer: We appreciated for your kind advice. In Figure 2.5C, the gene expression level of SPT1 was significantly reduced in TNFα-treated cells. HY7714 EPS treatment significantly increased even at the concentration of 0.1ug/mL, but there was no concentration-dependent effect. Also, we performed the western blot assay again for MMP13 expression level, and the results were corrected in Figure 6C.

6) In the discussion part line 22-23, authors mentioned that EPS can differ even among members of the same LAB strain, so how to detect and ensure the quality and purity of EPS if using it in cosmetic and functional dietary way?

Answer: Thanks for the suggestion. As the reviewer mentioned, it is very important to suggest the quality and purity of EPS in microorganisms. However, there is still limited research on the cultivation of microorganisms or industrial cultivation methods for the quality of EPS. Therefore, we have revised the contents of the discussion as follows.; “Given that microbial EPS has various effects of improving biological and cosmetic functions, it is very important to explore the biodiversity of the naturally derived LAB strains that produce high levels of EPS [5]. The yield of EPS synthesized by Lactobacill can be affected by composition of the medium and growth conditions [6]. In addition, EPS properties in slime form can negatively affect the gradual loss of probiotics, so efforts should be made to standardize culture methods to maintain quality and purity of EPS.”

[Reference]

  1. Sambuy, Y.; De Angelis, I.; Ranaldi, G.; Scarino, M.; Stammati, A.; Zucco, F. The Caco-2 cell line as a model of the intestinal barrier: influence of cell and culture-related factors on Caco-2 cell functional characteristics. Cell biology and toxicology 2005, 21, 1-26.
  2. van Breemen, R.B.; Li, Y. Caco-2 cell permeability assays to measure drug absorption. Expert opinion on drug metabolism & toxicology 2005, 1, 175-185.
  3. Patel, S.; Majumder, A.; Goyal, A. Potentials of exopolysaccharides from lactic acid bacteria. Indian journal of microbiology 2012, 52, 3-12.
  4. Lynch, K.M.; Coffey, A.; Arendt, E.K. Exopolysaccharide producing lactic acid bacteria: Their techno-functional role and potential application in gluten-free bread products. Food research international 2018, 110, 52-61.
  5. Korcz, E.; Varga, L. Exopolysaccharides from lactic acid bacteria: Techno-functional application in the food industry. Trends in Food Science & Technology 2021.
  6. Nguyen, P.-T.; Nguyen, T.-T.; Bui, D.-C.; Hong, P.-T.; Hoang, Q.-K.; Nguyen, H.-T. Exopolysaccharide production by lactic acid bacteria: the manipulation of environmental stresses for industrial applications. AIMS microbiology 2020, 6, 451.

Reviewer 2 Report

Comments for Molecules 1138823

This study aimed to evaluate whether certain HY7714 polysaccharides could serve as functional substances that act on the gut–skin axis to change the properties of dermal cells. The results showed the HY7714 EPS consists of a specific ratio of ribose, glucose, and mannose, and exhibits characteristics completely different from EPS in other types of L. plantarum. T. HY7714 EPS ameliorates the inflammatory response and MMP synthesis in Caco-2 cells, which are related to upregulation of MMPs following UVB exposure in HS68 skin fibroblasts, and thus affected regulation of the skin ECM. The authors suggest the potential of HY7714 EPS on against UVB-induced photoaging in human dermal cells and regulating tight junctions in human intestinal cells.

Comments

In general, this manuscript is adequately described, and the discussion and conclusions are partly supported by the data. However, the effects of HY7714 EPS on the protein levels of pro-inflammatory cytokines should be added in this study.

  1. HY7714 EPS should be clearly marked in the images of Fig 3 A-D exception of the concentration.
  2. The legend of Fig 5 “….. (A) TIMP1, (B) MMP1, (C) MMP3, (D) HAS1, (E) HAS2, and (F) SPT1” are wrong and different with the images.
  3. The authors showed the effects of HY7714 EPS on the relative mRNA levels of MMP1, MMP3, SPT1, HAS1, HAS2, and HAS3 on ECM modulation in UVB-irradiation HS68 cells in Fig 5, and the protein levels of MMPs and intracellular hyaluronic acid (HA) in in Fig 6. However, they only investigated the relative mRNA levels of pro-inflammatory cytokines in Fig 2 and Fig 7. The effects of HY7714 EPS on the protein levels of pro-inflammatory cytokines should be added in this study.

Author Response

Dear reviewer 2

Thank you for considering our manuscript for publication in Molecules. We are very pleasure to have been given the opportunity to revise our manuscript, “Exopolysaccharide from Lactobacillus plantarum HY7714 protects against skin aging through skin–gut axis communication”. We have addressed the reviewer’s comments point-by-point and made the necessary changes to the manuscript. We hope that the manuscript is now acceptable for publication in Molecules. Please see the attachment.

Sincerely,

Jung-Lyoul Lee, Ph.D

1) HY7714 EPS should be clearly marked in the images of Fig 3 A-D exception of the concentration.

Answer: Thank you for your kind advice. As reviewer’s comments, the name and concentration of the L. plantarum were clearly indicated in the images in Figure 3 A-D.

2) The legend of Fig 5 “….. (A) TIMP1, (B) MMP1, (C) MMP3, (D) HAS1, (E) HAS2, and (F) SPT1” are wrong and different with the images.

Answer: We appreciated pointing out our mistake. We revised it as follows.; “Figure 5. Effect of HY7714 EPS on ECM modulation in UVB-irradiation HS68 cells. Cells were irradiated with 30 mJ/m2 UVB and treated with or without HY7714 EPS. Relative mRNA levels of (A) MMP1, (B) MMP3, (C) SPT1, (D) HAS1, (E) HAS2, and (F) HAS3 in HS68 cells were monitored by qPCR and normalized against GAPDH. mRNA data are expressed as means ± SD (n = 4), and values labeled with different letters are significantly different, p < 0.05 (a > b > c > d).”

3) The authors showed the effects of HY7714 EPS on the relative mRNA levels of MMP1, MMP3, SPT1, HAS1, HAS2, and HAS3 on ECM modulation in UVB-irradiation HS68 cells in Fig 5, and the protein levels of MMPs and intracellular hyaluronic acid (HA) in in Fig 6. However, they only investigated the relative mRNA levels of pro-inflammatory cytokines in Fig 2 and Fig 7. The effects of HY7714 EPS on the protein levels of pro-inflammatory cytokines should be added in this study.

Answer: We appreciated for your recommendation. We absolutely agree that we should confirm the protein levels of cytokines. Therefore, considering the revision time, we have investigated the cytokine secretion level by using commercial ELISA kit. These data were added as Figure 7E-G and described in Result 2.7.; “Finally, secretion level of pro-inflammatory cytokines was investigated using ELISA kit. TNFα in HS68 cells increased after UVB exposure, but was slightly decreased by HY7714 EPS treatment but there was no significant difference (Figure 7E). While, IL-1β and IL-6 in HS68 cells increased after UVB exposure but were attenuated significantly by HY7714 EPS (Figure 7F-G). These results indicate that HY7714 EPS suppresses UVBS-induced release of pro-pro-inflammatory cytokines.”

Round 2

Reviewer 1 Report

 The writing is good and the figure/data are presented properly in a clear and concise form. The suggestions were considered by the authors and contributed further to scientific quality. Therefore, the manuscript was revised adequately and can be accepted for publication in Molecules

Reviewer 2 Report

The authors had corrected the mistakes and confirm the protein levels of cytokines by ELISA kit. These results indicate that HY7714 EPS suppresses UVBS-induced release of pro-pro-inflammatory cytokines. Furthermore, the authors also revised according to the opinion of other reviewers. Thus, suggested this manuscript could be accepted.